# The Mathematical Models of the Operation Process for Critical Production Facilities Using Advanced Technologies

**Vitaliy A. Yemelyanov [1], Anton A. Zhilenkov [2]** 📷**, Sergei G. Chernyi [2,3,4,\*]** 📷**, Anton Zinchenko [2] and Elena Zinchenko [2,4]**

1   Department of Business Informatics, Financial University under the Government of the Russian Federation, 125993 Moscow, Russia; Vaemelyanov@fa.ru
2   Department of Cyber-Physical Systems, Saint Petersburg State Marine Technical University, 198303 Saint-Petersburg, Russia; zhilenkovanton@gmail.com (A.A.Z.); Antel85@bk.ru (A.Z.); eltel85@bk.ru (E.Z.)
3   Department of Integrated Information Security, Admiral Makarov State University of Maritime and Inland Shipping, 198035 Saint-Petersburg, Russia
4   Department of Ship's Electrical Equipment and Automatization, Kerch State Maritime Technological University, 298309 Kerch, Russia
\*   Correspondence: sergiiblack@gmail.com

**Abstract:** The paper presents data on the problems of monitoring and diagnosing the technical conditions of critical production facilities, such as torpedo ladle cars, steel ladles. The accidents with critical production facilities, such as torpedo ladle cars, lead to losses and different types of damages in the metallurgical industry. The paper substantiates the need for a mathematical study of the operation process of the noted critical production facilities. A Markovian graph has been built that describes the states of torpedo ladle cars during their operation. A mathematical model is presented that allows determining the optimal frequency of diagnostics of torpedo ladle cars, which, in contrast to the existing approaches, take into account the procedures for preventive diagnostics of torpedo ladle cars, without taking them out of service. Dependence of the utilization coefficient on the period of diagnostics of PM350t torpedo ladle cars was developed. The results (of determining the optimal period of diagnostics for PM350t torpedo ladle cars) are demonstrated. The system for automated monitoring and diagnosing the technical conditions of torpedo ladle cars, without taking them out of service, has been developed and described.

**Keywords:** Markovian graph; operation process; torpedo ladle cars; diagnostic system; application software

## 1. Introduction

Throughout the entire metallurgical production cycle, tracking of hot metal and steel is constantly carried out to the blast furnace, converter, steelmaking, and other workshops. Hot metal and steel are tracked by a locomotive along the in-house railways using critical production facilities, namely torpedo ladle cars for transporting cast iron; ladles for steel transportation; and buckets for transporting cast iron. Torpedo ladle cars and steel ladles take charge of transferring molten iron and steel in iron and steel works, so they are key pieces of equipment in the metallurgical industry. As shown in paper [1], a ladle's lifetime influences production efficiency, product quality and flexibility, energy consumption, and the work conditions of workers.

High requirements are imposed on the operations and diagnostics of torpedo ladle cars associated with the influence of high temperatures, characteristic of hot metal (i.e., more than 1000 °C). High temperatures, which are exposed to torpedo ladle cars, cause the risk of destruction of such equipment, which is fraught with significant material losses and human casualties [2]. Due to the untimely diagnostics of the conditions of torpedo ladle

cars, the casing shell of the torpedo ladle car burns out, as a result of which, hot cast iron is poured onto the railroad bed. In turn, it damages the metallurgical plant from direct losses expressed in the cost of the torpedo ladle car, cast iron, and expenses for liquidation of consequences (restoration of the railway track). In addition, the metallurgical enterprise suffers indirect losses, expressed in the loss of profits due to delays in production caused by the restoration of railway communication within the production facilities.

In order to prevent accidents with such type of equipment, and to maintain industrial safety at the production facility, a growing number of diagnostic operations and technologies to control the technical conditions of lined equipment are applied in production units [3,4], which, in turn, require developments of new (and improvements of existing) technical means and information technologies.

It should be mentioned that modern automated systems are unable to provide a complete complex (qualitative and quantitative) automated evaluation of the lining conditions of the noted critical production facilities [5,6], which leads to a low level of objectiveness and quality of the decisions taken while exploiting the equipment. That is why scientific research—to create new systems and technologies for automated monitoring and diagnosing torpedo ladle cars and steel ladles—is relevant.

## 2. Related Works

Among the most significant works in the field of automation of diagnostics of the torpedo ladle cars and steel ladle state, performed earlier by other authors, one can single out papers [7–9], as well as software developments of companies such as "FLIR", "SICK", "PIEPER", and others [10–12]. Scientists in reference [13] developed recommendations for the automation of diagnostics of the torpedo ladle cars and steel ladle state. The insufficient level of automation—to diagnose the torpedo ladle cars of the iron and steel works—is highlighted as well.

To realize the maintenance procedures, based on the actual conditions of the steel ladles, leading manufacturers have developed computer diagnostic systems to monitor the operation processes and the current conditions of individual aggregates [14,15].

To perform the operations for maintenance of noted critical production facilities, special methods were developed by authors in [16], based on the use of the results from the thermographic measurements, in combination with those from mathematical models (describing the heat exchange processes and temperature fields, depending on the insulation thickness, type of refractory materials, and conditions of operation).

Authors in reference [17] introduced a laser profile meter and thermography at the ladle as evaluation methods, installed at Nippon Steel Corporation.

In paper [18], a two-dimensional mathematical model of a ladle based on the law of finite element method, was built, and the influence of thermal conductivity, thermal expansion coefficient, elasticity coefficient, and the thickness of work lining on the ladle stress field are discussed.

In paper [19], a multi-model approach was suggested to evaluate the defect criticality for the purpose of diagnostics of the condition, and making a decision on the maintenance and operation of the steel cast ladle. A decision support system for operation and maintenance of the steel casting ladles, for the purpose of safe utilization of the maximal resources, is presented and with using multifunctional instrumentation. During this period, instrumentation was developed with the use of modern computer vision systems, decision support, fuzzy systems, etc. Unfortunately, the high costs of devices seriously hinders their widespread distribution in all enterprises, but more enterprises can afford them. Many programs have been developed to support enterprises in the transition to new equipment.

Many ladle monitoring systems [10–12,20–25] automatically recognize predefined regions of interest of critical production facilities, and compare the measured temperatures with previously set parameters. However, the developed tools, models, and systems do not allow for the diagnoses of the noted critical production facilities without taking them out of service, and do not provide the opportunity for preventive diagnostics. Thus, there is

a need to improve automatic tools and information support to diagnose and monitor the technical states of the critical production facilities [25–27].

The main trend these days is towards "green steel production". The growing demand for high quality steel grades will require special attention to equipment in both new and modernized plants, and digitalization will be an integral part of all stages of production. Advanced metal manufacturing will be sustainable, carbon-free, safe, smart, modernized, and likely lead to materials with innovative properties and structures [28–30].

## 3. Mathematical Model of the Operation of Torpedo Ladle Cars

The purpose of the mathematical study, regarding the process of operation of torpedo ladle cars, is to build a model that describes the process of using torpedo ladle cars, which will allow determining the optimal frequency of their diagnostics. This need is due to the fact that the existing approaches [7–10] used to determine the frequency of diagnostics of torpedo ladle cars, based on the use of the normative value of the maximum permissible pouring of hot metal into the torpedo ladle car, have been outdated. The analysis of the papers and studies [1,5,13,22,23], showing untimely diagnostics of the conditions of torpedo ladle car, has led to known accidents, causing various types of damages to enterprises. The rapid development of technology, an increase in production capacity, a sharp increase in productivity, the struggle for the competitiveness of products, and the tightening of industrial and environmental safety requirements, have created the necessary prerequisites for the emergence and development of the direction of technical diagnostics.

In the model, the object of operation (a torpedo ladle car) is a set of its technical states, $S$, determined by the technological features of metallurgical production. Moreover, the process of the technical operations of a torpedo ladle car on its own can be defined as the process of the emergence and change of the operating modes of the torpedo ladle car in its various states, $S$, under the influence of certain external conditions.

The classical scheme of operation of the torpedo ladle cars can be represented as a set of states, in which they can be found:

$$S = \{s_1, s_2, s_3, s_4, s_5, s_6, s_7\} \tag{1}$$

where $s_1$—fully serviceable state of the torpedo ladle car (serviceable and/or in the mode of transportation of hot metal, standby mode, or decommissioning mode, due to the absence of the need for operation);

$s_2$—good condition of the torpedo ladle car, with signs of burnout of the lining;

$s_3$—the state of destruction of the torpedo ladle car (damaged or destroyed, and out of service);

$s_4$—state of routine diagnostics of the torpedo ladle car with decommissioning (decommissioning mode with determination of the state of the lining and casing shell);

$s_5$—state of repair of the torpedo ladle car auxiliary equipment (repair mode);

$s_6$—the state of repairs (replacement) of the lining (repair mode);

$s_7$—the state of the casing repairs (repair mode).

However, such a scheme has significant drawbacks, due to which the destruction of torpedo ladle cars occurs. The disadvantages are that, during such an operation (specific set of states), the torpedo ladle car is sent for technical diagnostics only after the resource of the nominal values of the operating parameters has expired (as a rule, this parameter is the number of maximum permissible pouring of hot metal into the torpedo ladle car). The problem is that, after replacing the lining, the torpedo ladle car does not withstand the nominal number of fillings, as a result of which, if the lining and the casing burnout is untimely detected, the torpedo ladle car is destroyed [27–30].

To solve this problem, it is proposed to build a model of the operation of a torpedo ladle car with the introduction of an additional state, which will allow for a preventive check of the technical condition of the torpedo ladle car without taking it out of service.

Let us build a Markovian model of a torpedo ladle car operation (Figure 1).

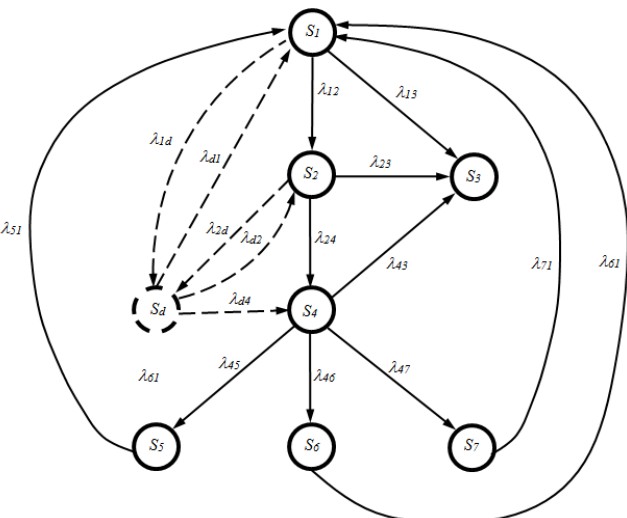

**Figure 1.** Markovian graph for the torpedo ladle car operation.

After preliminary research, a new state (dashed line in Figure 1) is added to the classical graph of the torpedo ladle car state (solid line in Figure 1): $s_d$—torpedo ladle car diagnostics state without decommissioning.

The new state of $s_d$ characterizes the mode of preventive maintenance of the torpedo ladle car, without taking it out of service. For the implementation and technical support of the introduced state, the authors in [24] created a method for determining the state of lined equipment by recognizing images of thermograms obtained by means of thermal imaging control [21].

Thus, we get a new set of states of the torpedo ladle car:

$$S' = \{s_1, s_2, s_3, s_4, s_5, s_6, s_7, s_d\} \tag{2}$$

Let us compose a system of Kolmogorov's differential equations for the original torpedo ladle car operation model (3) and the modified Markovian graph, taking into account the state $s_d$ (4).

$$\begin{cases} \frac{dP_{S_1}(t)}{dt} = -(\lambda_{12} + \lambda_{13})P_{S_1}(t) + \lambda_{51}P_{S_4}(t) + \lambda_{61}P_{S_6}(t) + \lambda_{71}P_{S_7}(t) \\ \frac{dP_{S_2}(t)}{dt} = -(\lambda_{23} + \lambda_{24})P_{S_2}(t) + \lambda_{12}P_{S_1}(t) \\ \frac{dP_{S_3}(t)}{dt} = \lambda_{13}P_{S_1}(t) + \lambda_{23}P_{S_2}(t) + \lambda_{43}P_{S_4}(t) \\ \frac{dP_{S_4}(t)}{dt} = -(\lambda_{43} + \lambda_{45} + \lambda_{46} + \lambda_{47})P_{S_4}(t) + \lambda_{24}P_{S_2}(t) \\ \frac{dP_{S_5}(t)}{dt} = -\lambda_{51}P_{S_5}(t) + \lambda_{45}P_{S_4}(t) \\ \frac{dP_{S_6}(t)}{dt} = -\lambda_{61}P_{S_6}(t) + \lambda_{46}P_{S_4}(t) \\ \frac{dP_{S_7}(t)}{dt} = -\lambda_{71}P_{S_7}(t) + \lambda_{47}P_{S_4}(t) \end{cases} \tag{3}$$

$$\begin{cases} \frac{dP_{S_1}(t)}{dt} = -(\lambda_{12} + \lambda_{1d} + \lambda_{13})P_{S_1}(t) + \lambda_{51}P_{S_4}(t) + \lambda_{61}P_{S_6}(t) + \lambda_{71}P_{S_7}(t) + \lambda_{d1}P_{S_d}(t) \\ \frac{dP_{S_2}(t)}{dt} = -(\lambda_{23} + \lambda_{24} + \lambda_{2d})P_{S_2}(t) + \lambda_{12}P_{S_1}(t) + \lambda_{d2}P_{S_d}(t) \\ \frac{dP_{S_3}(t)}{dt} = \lambda_{13}P_{S_1}(t) + \lambda_{23}P_{S_2}(t) + \lambda_{43}P_{S_4}(t) \\ \frac{dP_{S_4}(t)}{dt} = -(\lambda_{43} + \lambda_{45} + \lambda_{46} + \lambda_{47})P_{S_4}(t) + \lambda_{24}P_{S_2}(t) + \lambda_{d4}P_{S_d}(t) \\ \frac{dP_{S_5}(t)}{dt} = -\lambda_{51}P_{S_5}(t) + \lambda_{45}P_{S_4}(t) \\ \frac{dP_{S_6}(t)}{dt} = -\lambda_{61}P_{S_6}(t) + \lambda_{46}P_{S_4}(t) \\ \frac{dP_{S_7}(t)}{dt} = -\lambda_{71}P_{S_7}(t) + \lambda_{47}P_{S_4}(t) \\ \frac{dP_{S_d}(t)}{dt} = -(\lambda_{d1} + \lambda_{d2} + \lambda_{d4})P_{S_d}(t) + \lambda_{1d}P_{S_1}(t) + \lambda_{2d}P_{S_2}(t) \end{cases} \tag{4}$$

Thus, the system of differential Equation (4) is a mathematical model of the process of operation of torpedo ladle cars, taking into account their diagnostics without taking them out of normal operating modes.

The intensity of the transition is calculated based on real data on the transitions from state-to-state of torpedo ladle cars (of the PM350t type) used at Alchevsk Iron and Steel Works. The calculation is as follows:

$$\lambda_{ij} = \frac{n_{ij}^{type}}{n_i^{type} T_i^{type}} \tag{5}$$

where $n_i$—statistics on the number of torpedo ladle cars in the $i$-th state;

$n_{ij}$—the number of torpedo ladle cars that have passed from the $i$-th state to the $j$-th state in a fixed period $T_i$ (in this model, the $T_D$ value used in calculating the intensity of the transition to the $s_4$ state is of significant importance, since this value characterizes the frequency of the torpedo ladle car diagnostics);

*type*—the type of a torpedo ladle car (e.g., PM350t; MP600AS, etc.).

In this case, for modeling, the intervals $T_i$ must be of the same dimension, but not necessarily the same.

Depending on $\lambda_{ij}$ of the value $P_i(t)$, they can increase or decrease over time, except for the value $P_3(t)$, which can only increase, since state $s_3$ is absorbing and characterizes the destruction of the torpedo ladle car due to the burnout of its lining and casing shell without any possibility of restoration.

To determine the optimal period for diagnostics of $T_D$, let us consider the application of model (4) for a specific type of torpedo ladle cars (PM350t) used at the Alchevsk Iron and Steel Works. When constructing the Markovian model of readiness, statistical data on the operation of the PM350t torpedo ladle cars at Alchevsk Iron and Steel Works are used, reflecting the real values of the intensities of the transition from state-to-state.

Let us find the roots by solving the system of Equation (4) in the mathematical modeling environment Mathcad, using the built-in functions Given and Find (). To compare the basic and modified models of operating torpedo ladle cars, we will also solve system (3).

$$\textit{Given}$$
$$-(\lambda_{12} + \lambda_{1d} + \lambda_{13})P_{S_1} + \lambda_{51}P_{S_4} + \lambda_{61}P_{S_6} + \lambda_{71}P_{S_7} + \lambda_{d1}P_{S_d} = 0$$
$$-(\lambda_{23} + \lambda_{24} + \lambda_{2d})P_{S_2} + \lambda^{12}P_{S_1} + \lambda_{d2}P_{S_d} = 0$$
$$\lambda_{13}P_{S_1} + \lambda_{23}P_{S_2} + \lambda_{43}P_{S_4} = 0$$
$$-(\lambda_{43} + \lambda_{45} + \lambda_{46} + \lambda_{47})P_{S_4} + \lambda_{24}P_{S_2} + \lambda_{d4}P_{S_d} = 0$$
$$-\lambda_{51}P_{S_5} + \lambda_{45}P_{S_4} = 0 \tag{6}$$
$$-\lambda_{61}P_{S_6} + \lambda_{46}P_{S_4} = 0$$
$$-\lambda_{71}P_{S_7} + \lambda_{47}P_{S_4} = 0$$
$$-(\lambda_{d1} + \lambda_{d2} + \lambda_{d4})P_{S_d} + \lambda_{1d}P_{S_1} + \lambda_{2d}P_{S_2} = 0$$
$$P_{S_1} + P_{S_2} + P_{S_3} + P_{S_4} + P_{S_5} + P_{S_6} + P_{S_7} + P_{S_d} = 1$$

Finding the roots for the modified Markovian graph by means of Find (), we get:

$$Find\left(P_{S_1}, P_{S_2}, P_{S_3}, P_{S_4}, P_{S_5}, P_{S_6}, P_{S_7}, P_{S_d}\right) = \begin{pmatrix} 0,68 \\ 0,1 \\ 0,002 \\ 0,066 \\ 0,001 \\ 0,05 \\ 0,001 \\ 0,1 \end{pmatrix} \tag{7}$$

Find roots for a base graph:

$$Find\left(P_{S_1}', P_{S_2}', P_{S_3}', P_{S_4}, 'P_{S_5}', P_{S_6}', P_{S_7}'\right) = \begin{pmatrix} 0,593 \\ 0,010 \\ 0,076 \\ 0,189 \\ 0,001 \\ 0,127 \\ 0,004 \end{pmatrix} \tag{8}$$

The sum of probabilities $P_{S1} + P_{S12} + P_{Sd}$ characterizes the coefficient of technical use of a torpedo ladle car. Then the readiness function has the form:

$$K_{USAGE}(t) := P_{S_1}(t) + P_{S_2}(t) + P_{S_d}(t) \tag{9}$$

The optimal value can be obtained by plotting the dependence:

$$K_{USAGE} = f(T_D) \tag{10}$$

As a result of plotting the dependences of the technical utilization factor on the period of the PM350t diagnostic torpedo ladle cars, for the basic version and the modified version, the dependences shown in Figure 2 were obtained.

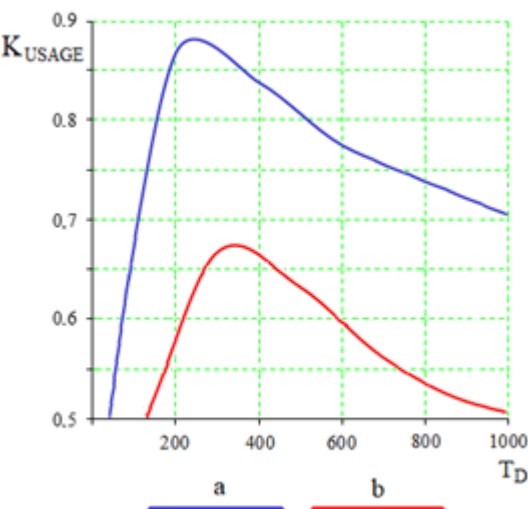

**Figure 2.** Dependences of the coefficients of technical use (Kusage) to determine the optimal diagnostic period for torpedo ladle cars: a—dependence of the technical utilization factor on the period of the PM350t diagnostic torpedo ladle cars for the modified version (with Sd); b—dependence of the technical utilization factor on the period of the PM350t diagnostic torpedo ladle cars for the basic version (without Sd).

Based on the developed dependences, the optimal period for diagnostics of the PM350t torpedo ladle cars was determined, which amounted to $T_D = 250$ ($T_D$—the period for diagnostics) pouring hot metal into the torpedo ladle car.

## 4. Development of the Diagnostic System to Determine the Technical State of the Torpedo Ladle Cars

The proposed mathematical model made it possible to create an automated system for diagnosing the torpedo ladle cars. A hydrogen-based direct reduction solution that allows direct use of any iron ore can virtually eliminate the carbon footprint of iron production. Carbon dioxide emissions will be close to zero. By-products will be recycled and processes carried out with maximum energy efficiency. Metallurgical giants, such as ArcelorMittal,

voestAlpine, SSAB, Dillinger, and a number of other manufacturers, have already begun to actively develop this technology. Japan's Nippon Steel has announced its intentions to abandon carbon technology in favor of hydrogen by 2025.

To implement the proposed model, the authors designed and developed a diagnostic system with specialized software. Figure 3 presents the structure of a diagnostic system with specialized software.

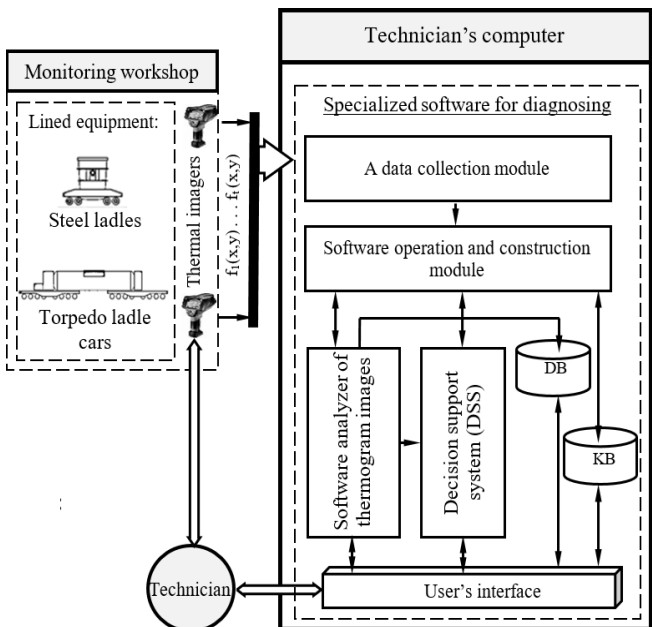

**Figure 3.** The structure of a diagnostic system with a specialized software.

According to the structure of a diagnostic system in Figure 3, it includes the following:

1. Thermal imagers to create thermogram images of a torpedo ladle car lining.
2. A computer with specialized software (Figure 4) to implement the neural network forecasting method [25], to assess the state of the PM350t torpedo ladle car. The specialized software consists of:

   - A module for data collection, designed to form initial data;
   - An operation module, designed to select methods and models for diagnostics and forecasting, to assess the state of PM350t torpedo ladle cars;
   - A thermogram image analyzer, designed to apply the intelligent methods of thermogram image processing, in order to diagnose and forecast the lining condition;
   - A decision support system (DSS) to operate knowledge in the process of technical diagnostics and forecasting, to assess the state of PM350t torpedo ladle cars;
   - Knowledge base (KB)—a storage of information that includes knowledge received after technical diagnostics and forecasting the state of PM350t torpedo ladle cars;
   - Database—storage of information that includes diagnostic operation data for different types of equipment.

Figure 4 presents the image-processing window of the developed specialized software to diagnose torpedo ladle cars.

Analyzing the existing experience in the development of the direction of technical diagnostics at these enterprises, three main "ways" can be distinguished:

- Selective episodic non-systemic diagnostics of pre-emergency equipment by attracting third-party experts or by our own small service.
- Periodic control of the entire fleet of equipment, according to the existing schedule, using portable control and measuring equipment by our own diagnostic service.

- Periodic or continuous monitoring of the entire fleet of equipment, according to the existing schedule, using a wide arsenal of external technical diagnostics (portable devices, stationary systems, bench complexes) by our own diagnostics service, numbering.

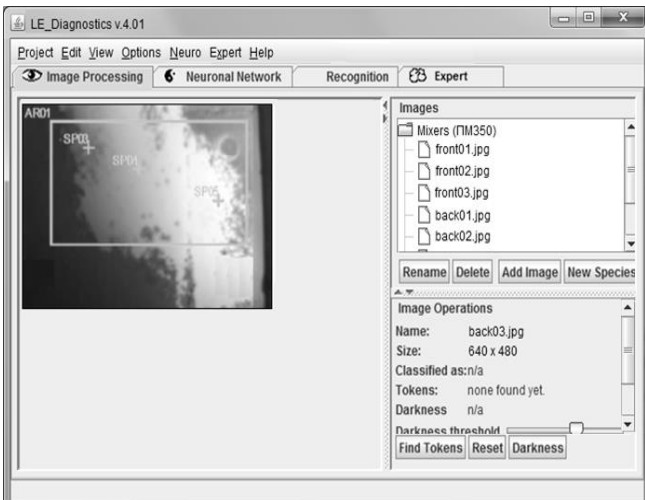

**Figure 4.** Developed specialized software to diagnose torpedo ladle cars (image processing window).

## 5. Discussion

Any modern metallurgical plant includes a large number of main and auxiliary industries, which operate a wide variety of equipment. Sudden failure of one of the units can be the reason for the disruption of the entire technological cycle; therefore, an important task is to organize monitoring of the conditions of the entire fleet of equipment to ensure a trouble-free operation. As practice has shown, due to a number of objective reasons, attracting engineering personnel of an enterprise to solve this problem is not always effective. The small number of engineering workers does not ensure the frequency of parameter controls necessary for trouble-free operation of the equipment. According to statistics, with the frequency of equipment inspections in accordance with the less frequent regulations, its overhaul interval increases the likelihood of missing a defect. In addition, the development of some defects, for example, the destruction of bearings, usually occurs in an "avalanche" manner. On the other hand, when diagnosing a large number of pieces of equipment, it is extremely difficult to collect the necessary additional information, such as temperature control, operating parameters, and to formalize the results of visual inspection.

In analysis mode, the functions of the developed specialized software are as follows:

- Input of the torpedo ladle car thermograms;
- Preliminary processing of the thermogram images;
- Forecasting the torpedo ladle car state by the approach [25];
- Recognizing the lining burnout zones by the approach [26];
- Evaluating the operational mode of torpedo ladle cars by the proposed mathematical model;
- Generating recommendations for the operation of torpedo ladle cars;
- Sending the results of diagnostics and forecasting the state of the torpedo ladle car to the workshop server.

In the training mode, the functions of the software are as follows:

- Inputting (by the technologist) the parameter values influencing the possibility of using the torpedo ladle car;
- Creating a neural network for thermogram recognition;
- Selecting a neural network architecture to forecast the state of the torpedo ladle car;
- Setting the parameters of the neural networks to forecast the torpedo ladle car state;
- Training the neural networks based on input data.

The developed software was implemented in the processes of the technical diagnostics, forecasting the state of PM350 torpedo ladle cars at Alchevsk Iron and Steel Works. The diagnostic system was developed as the following hardware configurations: the technician's computer (CPU—Intel Core i5 2.0 GHz; RAM—8 GB DDR3), the workshop server (CPU—Intel Xeon Gold 3.1 GHz; RAM—32 GB DDR4), thermal imagers FLIR GF309. Thermal specifications were: temperature range: 40 °C to 1500 °C; accuracy: $\pm 1$ °C for a temperature range of 0 °C to +100 °C or $\pm 2\%$ for a temperature range +100 °C; thermal sensitivity: <15 mK @ +30 °C.

In contrast to the existing approaches and systems described in [7,9–13,16,19,21], the developed system takes into account the procedure for preventive diagnostics of torpedo ladle cars. It provides the opportunity for diagnostics of torpedo ladle cars without taking them out of service.

The developed dependences in Figure 2 allow the optimal period for diagnostics of the PM350t torpedo ladle cars. According to the developed dependences in Figure 2, the optimal period for diagnostics of the PM350t torpedo ladle cars amounted to 250, pouring hot metal into the torpedo ladle car.

Based on the calculated number of the hot metal pouring and the number of "made" pouring in the torpedo ladle car, it becomes possible to estimate the remaining resources of the ladle car usage.

In the future, the proposed mathematical model can be effectively applied to determine the optimal period for diagnostics of the wide class of objects in the metallurgical industry, for example, for open cast iron trucks or steel ladle.

## 6. Conclusions

Special studies were conducted to check the effectiveness of the transfer of the system of maintenance and repair of industrial equipment to more advanced strategies, during which, various approaches to servicing different groups of units were analyzed using modern mathematical apparatus—Markov processes with discrete states and discrete time intervals. Research results have shown that the effectiveness of a particular form of maintenance directly depends on the cost of the equipment itself, its most vulnerable units, possible downtime, as well as the cost of current, scheduled, and emergency repairs.

Thus, the authors developed a mathematical model for the operation of torpedo ladle cars, which, in contrast to the existing approaches, takes into account the procedure for preventive diagnostics of torpedo ladle cars without taking them out of service. It makes it possible to expand a variety of maintenance strategies and diagnostics of torpedo ladle cars. The developed mathematical model of the process of operation of torpedo ladle cars allows determining the optimal frequency of diagnostics for this type of torpedo ladle car.

The diagnostic system was developed. It implements the methods of the proposed technology offered by the authors for automated monitoring of the technical conditions and supports decision-making in the operation of torpedo ladle cars.

**Author Contributions:** Conceptualization, S.G.C.; methodology V.A.Y., S.G.C., A.Z., A.A.Z., E.Z.; software V.A.Y., S.G.C., A.Z., E.Z.; validation V.A.Y., S.G.C., A.Z., E.Z.; formal analysis, S.G.C.; investigation V.A.Y., S.G.C., A.A.Z., E.Z.; resources V.A.Y., S.G.C., A.Z., E.Z.; data curation, S.G.C.; writing—original draft preparation, E.Z.; writing—review and editing, A.A.Z., S.G.C.; visualization, E.Z., A.Z.; supervision, V.A.Y.; project administration, S.G.C.; funding acquisition, S.G.C., E.Z., A.A.Z. All authors have read and agreed to the published version of the manuscript.

**Funding:** The research is partially funded by the Ministry of Science and Higher Education of the Russian Federation as part of World-Class Research Center program: Advanced Digital Technologies (contract no. 075-15-2020-903 dated 16 November 2020).

**Conflicts of Interest:** The authors declare no conflict of interest.

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
