# Peer review of "The Mathematical Models of the Operation Process for Critical Production Facilities Using Advanced Technologies"

_inventions, doi:10.3390/inventions7010008_

Round 1

Reviewer 1 Report

Thank you for providing me with the opportunity to read “The Mathematical Model of the Operation

Process of Critical Production Facilities”. I have the following comments:

  • The paper is not ready from a scientific point of view and needs a significant overhaul and major additions to be ready for publication. Please see my detailed comments below.
  • There is no mention of why this study needs to be conducted in the abstract. A very superficial and direct approach is adopted with no mention or relevance to any context for the study.
  • What are the benefits or practical contributions of the study? mention this in a line in the abstract
  • No coherence between the title and abstract such as the abstract doesn’t mention critical production facilities
  • The introduction is too short and doesn’t justify the paper as a scientific output. Please ass the following to the introduction:
    • The need to conduct this study
    • The problem to be solved
    • The potential contribution of the study
    • Any previous works conducted by others on this topic
    • How is the study organized?
  • Why is the f PM 350t torpedo ladle cars important to be studied? What is its significance? What benefits will this study bring? Justify these with academic references.
  • The paper needs a detailed literature section. This must clearly highlight the work conducted on this topic and review the state-of-the-art literature published in this domain. This must extend the references to at least 40. Currently, it is only 11, which are too low and highlights a lack of study on the authors’ behalf.
  • Add a proper method section to the paper and explain how the study is conducted, its various steps and justification of these steps.
  • The results and discussion sections are very weak and need serious reworks. The authors need to compare their results with other published works and highlight their innovation and how their model/study is superior to others.
  • What are the limitations of the study?
  • Provide the future direction for improvement of your work or exploring the topic further.

Author Response

Dear Reviewer 

Thank you for the corrections. We highlighted all the corrected fragments with color.

Why is the f PM 350t torpedo ladle cars important to be studied? What is its significance? What benefits will this study bring? Justify these with academic references.

Comments:        Introduction is rewritten:

                         Throughout the entire cycle of metallurgical production, tracking of hot metal and steel is constantly carried out to the blast furnace, converter, steelmaking and other workshops. Hot metal and steel are tracked by a locomotive along the in-house railways using critical production facilities, namely torpedo ladle cars for transporting cast iron; ladles for steel transportation; buckets for transporting cast iron. The torpedo ladle cars and steel ladles take charge of transferring molten iron and steel in iron and steel works, so it is a key equipment in metallurgical industry. As shown in paper [1] ladle’s lifetime influences production efficiency, product quality and flexibility, energy consumption, and work condition of workers.

High requirements are imposed on the operation and diagnostics of torpedo ladle cars associated with the influence of high temperatures characteristic of hot metal, more than 10000C. High temperatures, which are exposed to torpedo ladle cars, cause the risk of destruction of such equipment, which is fraught with significant material losses and human casualties [2]. Due to untimely diagnostics of the condition of torpedo ladle cars, the casing shell of the torpedo ladle car burns out, as a result of which hot cast iron is poured onto the railroad bed. In its turn, it damages the metallurgical plant from direct losses expressed in the cost of the torpedo ladle car, cast iron, and expenses for liquidation of consequences (restoration of the railway track). In addition, the metallurgical enterprise suffers indirect losses, expressed in the loss of profits due to delays in production caused by the restoration of railway communication within the production facilities.

In order to prevent accidents with such type of equipment and maintain industrial safety at the production facility, a growing number of diagnostic operations and technologies to control the technical condition of lined equipment are applied in production units [3, 4], which, in turn, require development of new and improvement of existing technical means and information technologies.

 It should be mentioned that modern automated systems are unable to provide a complete complex (qualitative and quantitative) automated evaluation of the lining condition of the noted critical production facilities [5, 6], which leads to a low level of objectiveness and quality of the decisions taken while exploiting the equipment. That is why scientific research to create new systems and technologies for automated monitoring and diagnosing torpedo ladle cars and steel ladles is relevant.

The paper needs a detailed literature section. This must clearly highlight the work conducted on this topic and review the state-of-the-art literature published in this domain. This must extend the references to at least 40. Currently, it is only 11, which are too low and highlights a lack of study on the authors’ behalf. Add a proper method section to the paper and explain how the study is conducted, its various steps and justification of these steps.

Comments: Added new paragraph in article and new references:

RELATED WORKS

Among the most significant works in the field of automation of diagnostics of the torpedo ladle cars and steel ladle state, performed earlier by other authors, one can single out papers [7, 8, 9], as well as a software developments of companies such as “FLIR”, “SICK”, “PIEPER” and others [10, 11, 12]. Scientists in the papers [13] have developed recommendations for the automation of diagnostics of the torpedo ladle cars and steel ladle state. The insufficient level of automation to diagnose the torpedo ladle cars of the iron and steel works is high-lighted as well.

To realize the maintenance procedures based on the actual condition of the steel ladles, leading manufacturers develop computer diagnostic systems to monitor the operation process and the current condition of the individual aggregates [14, 15]

To perform the operations for maintenance noted critical production facilities, special methods are developed by authors in [16] based on the use of the results from the thermographic measurements in combination with those from mathematical models (describing the heat exchange processes and temperature fields depending on the insulation thickness, type of refractory materials and conditions of operation).

Authors in research [17] introduce a laser profile meter and thermography at the ladle as evaluation methods installed at Nippon Steel Corporation.

In paper [18], two-dimensional mathematical model of ladle based on law of finite element method is built, and the influence of thermal conductivity, thermal expansion coefficient, elasticity coefficient and thickness of work lining on ladle stress field are discussed.

In paper [19], a multi-model approach has been suggested for evaluation of the defect criticality with the purpose of diagnostic of the condition and taking a decision on the maintenance and operation of the steel cast ladle. A Decision Support System for operation and maintenance of the steel casting ladles with the purpose of the safe utilization of their maximal resource is presented.

There are many ladle monitoring systems [10-12, 20, 21] that automatically recognize predefined regions of interest of critical production facilities and compare the measured temperature with previously set parameters. However, the developed tools, models, and systems do not allow the diagnosis of the noted critical production facilities without taking them out of service and do not provide the opportunity for preventive diagnostics. Thus, there is a need to improve automatic tools and information support for diagnosing and monitoring the technical state of the critical production facilities.

The results and discussion sections are very weak and need serious reworks. The authors need to compare their results with other published works and highlight their innovation and how their model/study is superior to others.

What are the limitations of the study?

Provide the future direction for improvement of your work or exploring the topic further.

Comments: Added new information in section «Discussion»:

In contrast to the existing approaches and systems described in [7, 9, 10-13, 16, 19, 21], the developed system takes into account the procedure for preventive diagnostics of torpedo ladle cars. It provides the opportunity for diagnostics of torpedo ladle cars without taking them out of service.

The developed dependences in Figure 2 allow the optimal period for diagnostics of the PM 350t torpedo ladle cars. According to the developed dependences in Figure 2 the optimal period for diagnostics of the PM 350t torpedo ladle cars is amounted to 250 pouring hot metal into the torpedo ladle car.

Based on the calculated number of the hot metal pouring and the number of made pouring in torpedo ladle car, it becomes possible to estimate the remaining resource of the ladle car usage. For instance, if the number of made pouring is 50 the remaining resource of the torpedo ladle car usage will be about 200 hot metal pouring.

The specification of the thermal imaging diagnostics for "torpedo ladle cars" has been omitted.

Comments: Added information about specification:

Thermal specification is: Temperature Range 40°C to 1,500°C, Accuracy ±1°C for temperature range 0°C to +100°C or ±2% of reading for temperature range +100°C, Thermal Sensitivity <15 mK @ +30°C.

Comments:        Abstract is rewritten:

The paper presents data on the problem of monitoring and diagnosing the technical condition of critical production facilities such as torpedo ladle cars, steel ladles. It is noted the accidents with critical production facilities like torpedo ladle cars lead to losses and different types of damage in metallurgical industry. The paper substantiates the need for a mathematical study of the operation process of the noted critical production facilities. A Markovian graph has been built that describes the states of torpedo ladle car cars during their operation. A mathematical model is presented that allows determining the optimal frequency of diagnostics of torpedo ladle cars, which, in contrast to the existing approaches, takes into account the procedure for preventive diagnostics of torpedo ladle cars without taking them out of service. The dependence of the coefficient of technical utilization on the period of diagnostics of torpedo torpedo ladle car cars PM350t is developed. The results of determining the optimal period of diagnostics for torpedo ladle car cars PM 350t are demonstrated. The system for automated monitoring and diagnosing the technical condition of torpedo ladle cars without taking them out of service has been developed and described.

Reviewer 2 Report

Interesting work, however there are some comments the authors should consider.

1) I would suggest not to mix Latin characters with Cyrillic ones (see p. 3-4). 

2) I think that it would be nice to present some relevant numerical data about the λij values used in the solution of (6)

3) Fig. 2 should contain the legend for the two curves.

Author Response

Dear Reviewer

Thank you for the corrections. We highlighted all the corrected fragments with color.

) I would suggest not to mix Latin characters with Cyrillic ones (see p. 3-4). 

correct 

2) I think that it would be nice to present some relevant numerical data about the λij values used in the solution of (6)

Corrected in the article

Depending on of the value, they can increase or decrease over time, except for the value, which can only increase, since the state s3 is absorbing and characterizes the destruction of the torpedo ladle car due to the burnout of its lining and casing shell without any possibility of restoration.

To determine the optimal period for diagnostics of ТD, let us consider the application of model (4) for a specific type of torpedo ladle cars PM350t used at the Alchevsk Iron and Steel Works. When constructing the Markovian model of readiness, statistical data on the operation of the PM350t torpedo ladle cars at the Alchevsk Iron and Steel Works are used, reflecting the real values of the intensities of the transition from state to state.

Let us find the roots by solving the system of equations (4) in the mathematical modeling environment MathCAD using the built-in functions Given and Find(). To compare the basic and modified models of operating torpedo ladle cars, we will also solve system (3).

3) Fig. 2 should contain the legend for the two curves.

FIGURE 2. Dependences of the coefficient of technical use to determine the optimal diagnostic period for torpedo ladle cars:

a - dependence of the technical utilization factor on the period of the PM 350t diagnostic torpedo ladle cars for the modified version (with Sd);

b - dependence of the technical utilization factor on the period of the PM 350t diagnostic torpedo ladle cars for the basic version (without Sd).

On the basis of the developed dependences, the optimal period for diagnostics of the PM 350t torpedo ladle cars was determined, which amounted to  pouring hot metal into the torpedo ladle car.

Reviewer 3 Report

According to the authors' statement, the central issue of the paper is "To determine the optimal period for diagnostics of ... torpedo ladle cars PM350t ...". Input data for "torpedo ladle cars" was obtained from thermography images. The mathematical model of the operation of torpedo ladle cars proposed in the work is based on the Markovian graph. After reading the paper, a doubt arises because it is difficult to identify the novelty. The specification of the thermal imaging diagnostics for "torpedo ladle cars" has been omitted.

Author Response

Dear Editor

We add information to introduction part

I. INTRODUCTION

Throughout the entire cycle of metallurgical production, tracking of hot metal and steel is constantly carried out to the blast furnace, converter, steelmaking and other workshops. Hot metal and steel are tracked by a locomotive along the in-house railways using critical production facilities, namely torpedo ladle cars for transporting cast iron; ladles for steel transportation; buckets for transporting cast iron. The torpedo ladle cars and steel ladles take charge of transferring molten iron and steel in iron and steel works, so it is a key equipment in metallurgical industry. As shown in paper [1] ladle’s lifetime influences production efficiency, product quality and flexibility, energy consumption, and work condition of workers.

High requirements are imposed on the operation and diagnostics of torpedo ladle cars associated with the influence of high temperatures characteristic of hot metal, more than 10000C. High temperatures, which are exposed to torpedo ladle cars, cause the risk of destruction of such equipment, which is fraught with significant material losses and human casualties [2]. Due to untimely diagnostics of the condition of torpedo ladle cars, the casing shell of the torpedo ladle car burns out, as a result of which hot cast iron is poured onto the railroad bed. In its turn, it damages the metallurgical plant from direct losses expressed in the cost of the torpedo ladle car, cast iron, and expenses for liquidation of consequences (restoration of the railway track). In addition, the metallurgical enterprise suffers indirect losses, expressed in the loss of profits due to delays in production caused by the restoration of railway communication within the production facilities.

In order to prevent accidents with such type of equipment and maintain industrial safety at the production facility, a growing number of diagnostic operations and technologies to control the technical condition of lined equipment are applied in production units [3, 4], which, in turn, require development of new and improvement of existing technical means and information technologies.

 It should be mentioned that modern automated systems are unable to provide a complete complex (qualitative and quantitative) automated evaluation of the lining condition of the noted critical production facilities [5, 6], which leads to a low level of objectiveness and quality of the decisions taken while exploiting the equipment. That is why scientific research to create new systems and technologies for automated monitoring and diagnosing torpedo ladle cars and steel ladles is relevant

We corrected two parts with new information and your comments

II. RELATED WORKS

Among the most significant works in the field of automation of diagnostics of the torpedo ladle cars and steel ladle state, performed earlier by other authors, one can single out papers [7, 8, 9], as well as a software developments of companies such as “FLIR”, “SICK”, “PIEPER” and others [10, 11, 12]. Scientists in the papers [13] have developed recommendations for the automation of diagnostics of the torpedo ladle cars and steel ladle state. The insufficient level of automation to diagnose the torpedo ladle cars of the iron and steel works is high-lighted as well.

To realize the maintenance procedures based on the actual condition of the steel ladles, leading manufacturers develop computer diagnostic systems to monitor the operation process and the current condition of the individual aggregates [14, 15]

To perform the operations for maintenance noted critical production facilities, special methods are developed by authors in [16] based on the use of the results from the thermographic measurements in combination with those from mathematical models (describing the heat exchange processes and temperature fields depending on the insulation thickness, type of refractory materials and conditions of operation).

Authors in research [17] introduce a laser profile meter and thermography at the ladle as evaluation methods installed at Nippon Steel Corporation.

In paper [18], two-dimensional mathematical model of ladle based on law of finite element method is built, and the influence of thermal conductivity, thermal expansion coefficient, elasticity coefficient and thickness of work lining on ladle stress field are discussed.

In paper [19], a multi-model approach has been suggested for evaluation of the defect criticality with the purpose of diagnostic of the condition and taking a decision on the maintenance and operation of the steel cast ladle. A Decision Support System for operation and maintenance of the steel casting ladles with the purpose of the safe utilization of their maximal resource is presented.

There are many ladle monitoring systems [10-12, 20, 21] that automatically recognize predefined regions of interest of critical production facilities and compare the measured temperature with previously set parameters. However, the developed tools, models, and systems do not allow the diagnosis of the noted critical production facilities without taking them out of service and do not provide the opportunity for preventive diagnostics. Thus, there is a need to improve automatic tools and information support for diagnosing and monitoring the technical state of the critical production facilities.

III. MATHEMATICAL MODEL OF THE OPERATION OF TORPEDO LADLE CARS

The purpose of the mathematical study of the process of operation of torpedo ladle cars is to build a model that describes the process of using torpedo ladle cars, which will allow determining the optimal frequency of their diagnostics. This need is due to the fact that the existing approaches [7-10] to determining the frequency of diagnostics of torpedo ladle cars, based on the use of the normative value of the maximum permissible pouring of hot metal into the torpedo ladle car, have been outdated. The analysis of the papers and studies [1, 5, 13, 22, 23] shown untimely diagnostics of the condition of torpedo ladle cars, leads to known accidents causing various types of damage to enterprises.

In the model, the object of operation (a torpedo ladle car) is a set of its technical states S, determined by the technological features of metallurgical production. Moreover, the process of technical operation of a torpedo ladle car on its own can be defined as the process of the emergence and change of operating modes of the torpedo ladle car in its various states S under the influence of certain external conditions.

The classical scheme of operation of torpedo ladle cars  can be represented as a set of states in which they can be found/

We add new references and information for part disscusion 

Thermal specification is: Temperature Range 40°C to 1,500°C, Accuracy ±1°C for temperature range 0°C to +100°C or ±2% of reading for temperature range +100°C, Thermal Sensitivity <15 mK @ +30°C.

In contrast to the existing approaches and systems described in [7, 9, 10-13, 16, 19, 21], the developed system takes into account the procedure for preventive diagnostics of torpedo ladle cars. It provides the opportunity for diagnostics of torpedo ladle cars without taking them out of service.

The developed dependences in Figure 2 allow the optimal period for diagnostics of the PM 350t torpedo ladle cars. According to the developed dependences in Figure 2 the optimal period for diagnostics of the PM 350t torpedo ladle cars is amounted to 250 pouring hot metal into the torpedo ladle car.

Based on the calculated number of the hot metal pouring and the number of made pouring in torpedo ladle car, it becomes possible to estimate the remaining resource of the ladle car usage. For instance, if the number of made pouring is 50 the remaining resource of the torpedo ladle car usage will be about 200 hot metal pouring.

Thank you for the corrections. We highlighted all the corrected fragments with color.

Round 2

Reviewer 3 Report

I'm still having trouble pinpointing the  research thread novelty. However, I believe it can be published.

A minor remark
Fig. 2 and text: TD - no units

Author Response

Dear Reviewer. We took into account your comments and expanded our article with new data on the research problem.
Figure 2 changed the name and gave a decoding of the coefficient (TD -  the period for diagnostics 

Figure 2 -  Dependences of the coefficient of technical use (Kusage) to determine the optimal diagnostic period for torpedo ladle cars

).
We presented the relevance of the research problem and added additional links to publications on the problem where the prospect of the research and its importance are discussed.

the title has been corrected. New text is highlighted in the publication. the full affiliation of the authors is indicated.
I hope this version will be of interest to you.
Thank you for working with our article.